# Extracellular-matrix-mediated osmotic pressure drives *Vibrio cholerae* biofilm expansion and cheater exclusion

Jing Yan[1,2], Carey D. Nadell[1,3], Howard A. Stone [2], Ned S. Wingreen[1] & Bonnie L. Bassler [1,4]

Biofilms, surface-attached communities of bacteria encased in an extracellular matrix, are a major mode of bacterial life. How the material properties of the matrix contribute to biofilm growth and robustness is largely unexplored, in particular in response to environmental perturbations such as changes in osmotic pressure. Here, using *Vibrio cholerae* as our model organism, we show that during active cell growth, matrix production enables biofilm-dwelling bacterial cells to establish an osmotic pressure difference between the biofilm and the external environment. This pressure difference promotes biofilm expansion on nutritious surfaces by physically swelling the colony, which enhances nutrient uptake, and enables matrix-producing cells to outcompete non-matrix-producing cheaters via physical exclusion. Osmotic pressure together with crosslinking of the matrix also controls the growth of submerged biofilms and their susceptibility to invasion by planktonic cells. As the basic physicochemical principles of matrix crosslinking and osmotic swelling are universal, our findings may have implications for other biofilm-forming bacterial species.

[1] Department of Molecular Biology, Princeton University, Princeton, NJ 08544, USA. [2] Department of Mechanical and Aerospace Engineering, Princeton University, Princeton, NJ 08544, USA. [3] Max Planck Institute for Terrestrial Microbiology, 35043 Marburg, Germany. [4] Howard Hughes Medical Institute, Chevy Chase, MD 20815, USA. Correspondence and requests for materials should be addressed to N.S.W. (email: wingreen@princeton.edu) or to B.L.B. (email: bbassler@princeton.edu)

Bacteria survive over a remarkable range of osmotic pressures[1]. Indeed, some bacteria can transition between fresh water and sea water, withstanding a change in osmotic pressure up to 50 atm (~2 Osm)[2]. Adaptation to extremes in osmolarity depends on active and passive mechanisms that maintain constant osmotic pressure differentials between individual cells and the environment[1, 2]. However, how bacteria respond collectively to osmotic pressure changes is not clear, particularly in spatially structured communities such as biofilms[3, 4]—surface-attached bacterial collectives embedded in a secreted polymeric matrix[5]. Biofilms are an underlying source of chronic infection[6], and clog networks and filters in industry[7], but they are also useful in contexts such as waste-water treatment[8] and microbial fuel cells[9].

The biofilm matrix protects the embedded cells against environmental insults such as mechanical shear, predation, invasion, and antibiotics[5, 10]. Major components of the typical matrix are extracellular polysaccharides (EPS), which function in conjunction with accessory proteins and, in some cases, extracellular DNA[11]. Intensive research has focused on defining the functions of the matrix components and the regulatory mechanisms driving matrix production[12]. Much less well studied are the physical nature and material properties of matrix

networks[11, 13]. In biofilms, the high local concentration of polymer molecules surrounding the cells necessarily produces an osmotic pressure difference between the matrix and the external environment[14]. This pressure differential is likely an important environmental parameter that varies depending on context, for example, from hypotonic fresh water to saline ocean water to hyper-saline sludge environments. The influence of osmotic pressure gradients on the growth characteristics of biofilms and the fitness of the bacteria residing in them remains under-explored beyond a few seminal studies[14–17].

Pioneering work by Seminara et al. analyzing *Bacillus subtilis* colonies on agar plates suggested a crucial role for EPS-generated osmotic pressure differences in facilitating nutrient uptake[14]. Specifically, matrix-secreting colonies of *B. subtilis* expanded more rapidly than colonies of non-matrix-secreting cells, leading Seminara et al. to develop a theory for water transport into biofilms in which the biofilms were modeled as a viscous fluid with secreted EPS modeled as an extracellular osmolyte. Taking this precedent as motivation, we investigate the generality of osmotic-pressure-driven expansion of biofilms on air-solid interfaces as well as on submerged surfaces, using a different model bacterial biofilm producer, *Vibrio cholerae*[18], the causal agent of the pandemic disease cholera. We systematically validate

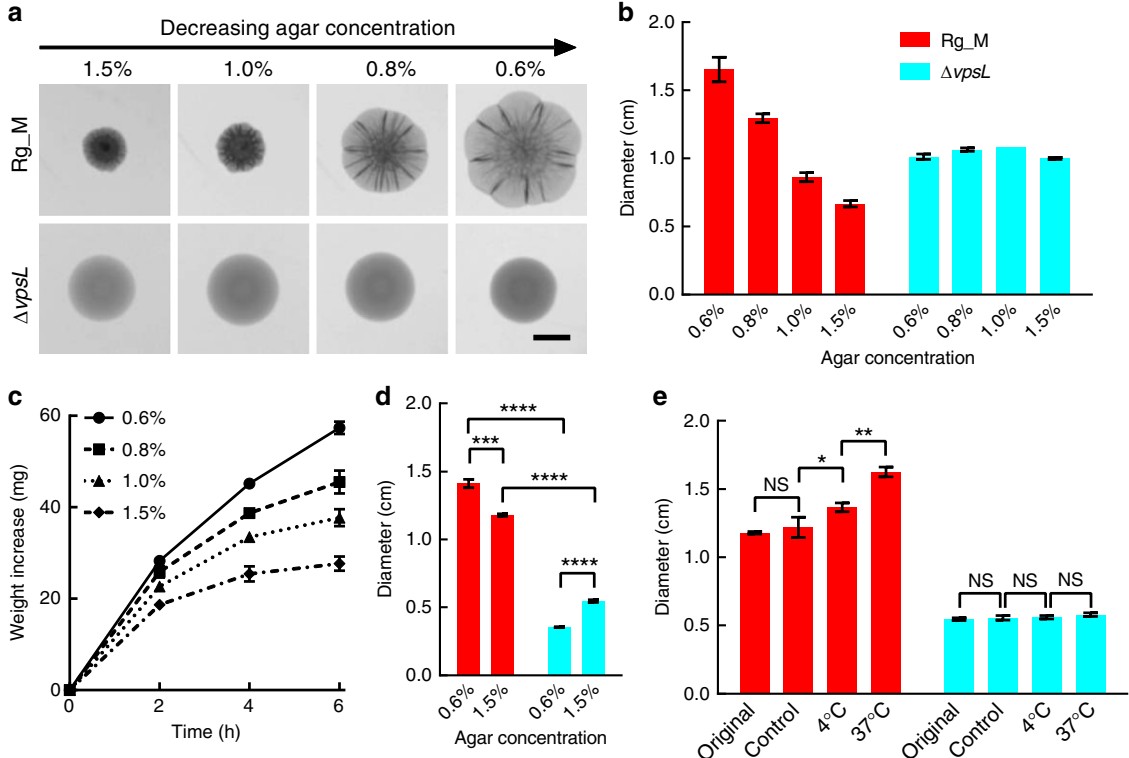

**Fig. 1** Osmotic pressure drives *V. cholerae* colony expansion. **a** Representative images of colony biofilms of the rugose (Rg_M) and EPS⁻ (Δ*vpsL*) *V. cholerae* strains grown for 2 days on LB medium containing the designated percentages of agar. *Scale bar*: 0.5 cm. **b** Colony biofilm diameter as a function of agar concentration (rugose (Rg_M: *red*) and EPS⁻ (Δ*vpsL*: *cyan*)). **c** To mimic the swelling effect that occurs in colonies exposed to osmotic pressure changes, 20 μL liquid droplets of LB containing 15% dextran (~2000 kDa) were placed on semipermeable membranes on top of LB medium solidified with the designated concentrations of agar. Increases in the droplet weight were measured over time. **d** Diameters of 4-day-old colony biofilms of Rg_M (*red*) and Δ*vpsL* (*cyan*) strains grown on semipermeable membranes placed on LB medium solidified with the specified percentages of agar. Unpaired *t*-tests with Welch's correction were performed for statistical analyses. **e** To differentiate the contribution of growth from that of passive swelling during colony expansion, Rg_M (*red*) and Δ*vpsL* (*cyan*) colonies were initially grown for 4 days on semipermeable membranes on top of LB medium solidified by 1.5% agar at 37 °C (denoted Original). The membranes containing the colonies were subsequently floated on LB liquid medium overnight at 4 or at 37 °C. To control for growth, a separate set of plates containing membranes with colonies that had been grown for 4 days were held at 4 °C overnight (denoted Control). In all cases, the diameters of the colonies were measured after 20 h. Paired *t*-tests were performed for statistical analyses. NS denotes not significant; *denotes $P < 0.05$, **denotes $P < 0.01$, ***denotes $P < 0.001$, and ****denotes $P < 0.0001$. *Error bars* correspond to standard deviations with $n = 3$ in **c** and $n = 4$ in other panels

the importance of the osmotic pressure gradient in driving colony expansion. We also examine the differences in biofilm growth and expansion that arise due to the hydrogel-like elastic properties of the *V. cholerae* biofilm matrix. In particular, we assess the consequences of osmotic pressure differentials on colony morphologies and we characterize the individual and combined contributions of particular extracellular matrix protein components to osmotic expansion. Finally, we explore the ecological consequences of osmotic pressure on biofilm-producing cells.

## Results

**Osmotic pressure changes drive colony biofilm expansion**. To explore the physical principles connecting osmotic pressure differentials to colony biofilm growth, we studied a commonly used *V. cholerae* constitutive biofilm-forming strain[19–21]. This strain has a missense mutation ($vpvC^{W240R}$) that causes elevated levels of the second messenger molecule c-di-GMP which, in turn, drives hypersecretion of the biofilm matrix[19]. This strain is denoted Rg for "rugose" and was used as the parent strain in all of the submerged biofilm analyses. To avoid the confounding effects of *V. cholerae* swarming on semi-solid media, we deleted the flagellar motor gene $pomA$[22] in the $vpvC^{W240R}$ mutant background. The resulting strain, which is denoted Rg_M, is the parent strain used for all of the colony biofilm analyses. All additional mutations and fluorescent reporter fusions were introduced into these two parent strains.

Figure 1a shows representative transmission microscopy images of 2-day-old *V. cholerae* Rg_M colony biofilms grown at 37 °C on Luria-Bertani (LB) medium solidified with different percentages of agar. The Rg_M parent (*top* row) shows a dramatic increase in colony size as a function of decreasing agar concentration, as quantified in Fig. 1b. For example, the colony diameter on a 0.6% agar plate is ~3 times larger than that on a 1.5% agar plate. In contrast, the sizes of colonies of an EPS⁻ mutant (Δ*vpsL*; a gene required for synthesis of the key matrix polysaccharide)[23] do not depend on the agar concentration. These findings demonstrate that EPS production is essential for *V. cholerae* colony biofilms to expand differentially in response to changes in agar concentration.

We hypothesize that the differences in colony size on media containing various concentrations of agar stem from the different osmotic pressure contrasts across the colony-agar interfaces during active cell growth. This inference mirrors that of Seminara et al.[14] from their studies of the Gram-positive biofilm former *B. subtilis*. The internal osmotic pressure of the biofilms is primarily generated by the EPS, a macromolecular osmolyte secreted and maintained by the cells. The resulting interfacial osmotic pressure contrast could boost colony biofilm expansion by drawing nutrient-carrying flow through the interface, thereby promoting growth and at the same time physically swelling the entire colony. To test this physical picture, we devised an experiment to mimic features of the process occurring in Fig. 1a. We placed liquid droplets containing the macromolecular osmolyte dextran (15%: molecular weight ~2000 kDa) on semipermeable membranes on top of agar plates containing different percentages of agar (Fig. 1c and Supplementary Fig. 1). The semipermeable membranes prevent the dextran polymers from diffusing into the agar, simulating the EPS surrounding and attached to biofilm cells. However, liquid can pass through the membrane, and the direction and speed of the flow will depend on the osmotic pressure difference. Indeed, uptake of water from the agar plate across the membrane and into the dextran solution occurred on the time scale of a few hours. Moreover, lower percentages of agar generated higher osmotic pressure differences

across the interfaces, and hence, increased water uptake by the droplet. The results from this model experiment parallel the colony biofilm size measurements (Fig. 1a, b). Hence, we find that biofilm matrix-producing cells could employ the same physical driving force, i.e., the osmotic pressure contrast, to expand their colony biofilms.

An alternative or additional mechanism that could contribute to the observed dependence of biofilm colony size on agar concentration concerns the change in agar stiffness that occurs as a function of agar concentration[24]. To separate mechanical contributions stemming from agar stiffness from contributions arising from osmotic pressure on colony growth, we grew *V. cholerae* colony biofilms on semipermeable membranes atop different percentage agar plates (Fig. 1d). In this arrangement, because the bacteria are not in direct contact with the agar, the only differential interaction they experience is the osmotic-contrast driving force that depends on the underlying agar concentration. Colony biofilm size continued to be negatively correlated with the underlying agar concentration in the Rg_M strain, verifying that the osmotic pressure contrast plays a role in determining colony biofilm size. We also note that the overall colonies expanded much slower when on membranes than when grown directly on the agar, presumably due to the increased friction colony biofilms experience on the semipermeable membrane surface compared to on agar. Notably, colony biofilm growth of the Δ*vpsL* strain slowed to a much larger extent than did growth of the Rg_M strain on the membrane compared to on agar: the Δ*vpsL* colony biofilms were two–three times smaller in diameter than the Rg_M colony biofilms, irrespective of the agar concentration. We hypothesize that the spreading of the droplet-like colonies of the Δ*vpsL* strain was retarded more on the semipermeable membrane than on agar because of the membrane's reduced wettability (shown by its larger contact angle in Supplementary Fig. 2)[16]. We can compare these results to those in Fig. 1b, in which the size of the Rg_M colony biofilm is larger than that of the Δ*vpsL* strain on low-concentration agar but smaller than the Δ*vpsL* strain on high-concentration agar, with a crossover agar concentration between 0.8 and 1.0%. On agar plates, the presence of the polymeric matrix hampers the expansion of the Rg_M colony biofilm when the osmotic differential is low. With decreasing agar concentration, increased Rg_M cell growth and biofilm expansion occur because the osmotic differential dominates the growth/expansion process. We note that such a crossover in relative size is not observed in the case of *B. subtilis* biofilms[14], in which the matrix-producing colonies are consistently larger than non-producing colonies, suggesting a fundamental difference in matrix properties between the two species, which we investigate further in later sections.

The above results demonstrate a role for osmotic pressure in assisting *V. cholerae* colony biofilm expansion, and suggest two interdependent mechanisms underlying expansion: (1) physical swelling of the entire colony biofilm and (2) enhanced cell division due to increased influx of nutrients. To prove that both phenomena are relevant and to quantify the respective contributions of the two mechanisms, we first grew *V. cholerae* Rg_M colony biofilms on semipermeable membranes on top of 1.5% agar plates. Subsequently, each membrane containing a colony biofilm was transferred to the surface of liquid LB medium, which generates an even larger osmotic pressure imbalance than does the agar plate. We compared the colony biofilm sizes before and after overnight flotation on the liquid medium. We performed the experiment at both 4 and 37 °C to control for the cell growth rate (Fig. 1e). Control experiments show that cell doublings at 4 °C were negligible. Nonetheless, at 4 °C, the Rg_M colony biofilm diameter expanded by 16% following exposure to the increased osmotic pressure contrast.

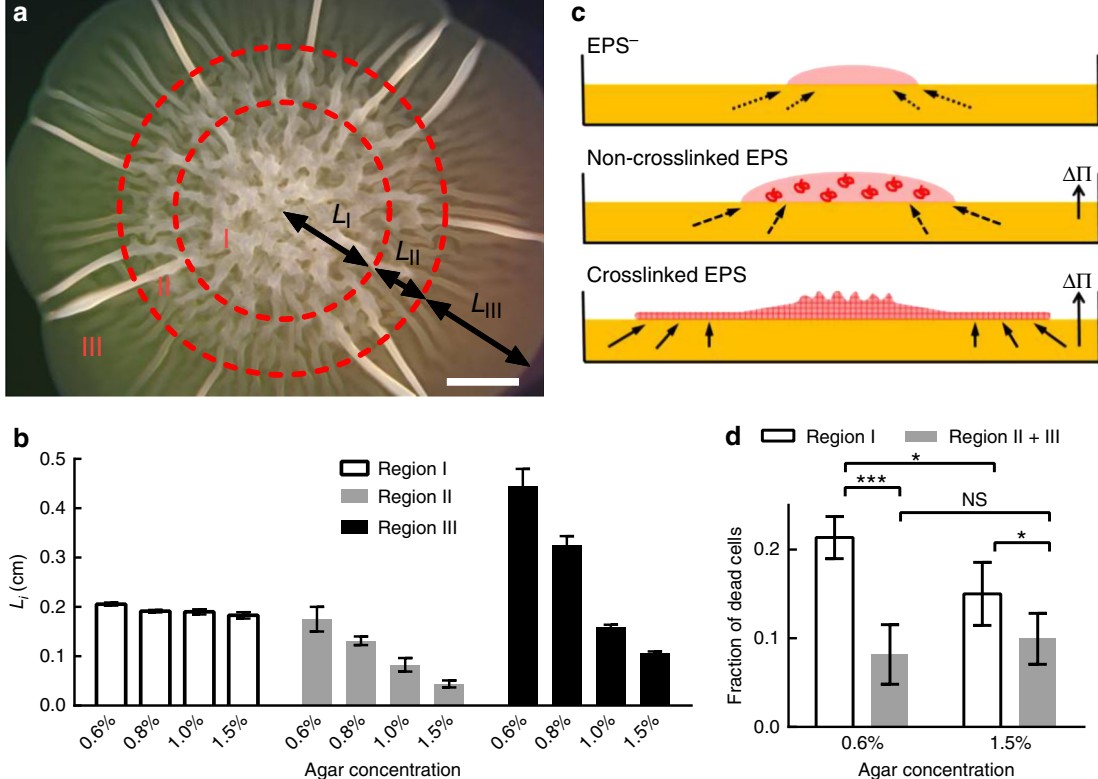

**Fig. 2** *V. cholerae* colony biofilms display distinct spatially patterned regions that depend on osmotic pressure. **a** Representative 2-day-old Rg_M colony grown on LB medium containing 0.6% agar imaged by stereomicroscopy. Three regions can be identified (denoted I, II, and III). $L_i$ denotes the length scale of each reproducibly identifiable region. *Scale bar*: 0.2 cm. **b** Quantitation of $L_i$ as a function of agar concentration. **c** Schematic representation of the proposed colony expansion mechanism for the EPS⁻ (*top*) strain, a strain containing non-crosslinked EPS (*middle*), and a strain containing crosslinked EPS (*bottom*). *Pink* denotes the bacterial colony (and matrix if present). *Yellow* denotes the solid surface, which contains nutrients. The *red coils* (*middle*) denote non-crosslinked matrix polymers. ΔΠ denotes the osmotic pressure difference across the interface between the colony and the solid surface. The *red mesh* (*bottom*) depicts crosslinked matrix that forms a hydrogel network. *Arrows* with *dotted lines* denote nutrient uptake by diffusion (*top*); *solid arrows* denote nutrient transport due to the osmotic pressure difference (*bottom*), and *arrows* with *dashed lines* denote the intermediate case (*middle*). **d** Region I is the nutrient-limited zone in which higher cell death occurs compared to regions II and III. Paired *t*-tests were performed in **d** within the groups and unpaired *t*-tests with Welch's correction were performed between the groups. NS denotes not significant; *denotes $P < 0.05$, and ***denotes $P < 0.001$. All *error bars* correspond to standard deviations with $n = 4$

Therefore, physical swelling does occur. Moreover, the colony biofilm diameter increased by a total of 38% if growth was allowed (37 °C, Fig. 1e). Finally, neither of these colony biofilm expansion mechanisms occurred in colonies lacking EPS (Fig. 1e). We conclude that both passive physical swelling of the colony matrix and enhanced nutrient uptake leading to increased cell doubling contribute to *V. cholerae* colony biofilm expansion, and both processes are only possible if the cells produce EPS, which functions as an extracellular osmolyte.

**Osmotic pressure drives biofilm spatial architecture.** Bacteria form distinct colony architectures on agar plates, reflecting different underlying growth and patterning programs[25, 26]. We therefore investigated what changes occur in the colony biofilm architecture in response to osmotic-pressure-driven expansion. *V. cholerae* Rg_M colony biofilms contain three identifiable regions (Fig. 2a). Region I is located at the center and possesses randomly oriented ridges. Region II is a transition zone of decreasing biofilm thickness. The most interesting area is region III, at the outer-most zone, featuring occasional, radially orientated ridges extending to the periphery of the colony biofilm. In between the ridges, the colony is essentially flat (i.e., two dimensional), as judged from the homogeneous intensity in the transmission image (Fig. 1a). Region I shows no size dependence

on agar concentration, whereas both regions II and III show strong size dependence (Fig. 2b). Δ*vpsL* mutant colonies lack these visible architectural features and appear as homogeneous droplets (Fig. 1a).

We propose a model, based on the above data, to account for these three regions (Fig. 2c). For bacterial cells that are unable to produce EPS, expansion occurs exclusively through cell division[14], and nutrients enter the colony biofilm via diffusion (Fig. 2c, *top*)[27]. Over time, the cells at the core become depleted of nutrients, and therefore only cells at the edge of the community grow[28]. In colony biofilms that possess non-crosslinked or weakly crosslinked EPS (Fig. 2c, *middle* and discussed further in the next section), an osmotic pressure difference across the biofilm-substratum interface enhances nutrient uptake; however, the overall colony biofilm remains featureless, with a smooth 3D contour. By contrast, the *V. cholerae* Rg_M colony biofilm expands in 2D, coupled with ridge formation characteristic of a thin elastic material[29, 30]. These observations are consistent with the reported elastic properties of *V. cholerae* biofilms[31]. We therefore suggest that the *V. cholerae* biofilm matrix is similar to a crosslinked hydrogel (Fig. 2c, *bottom*)[13, 32]. Within this context, we can rationalize how the morphological features of the *V. cholerae* Rg_M colony biofilm arise. Region I is the nutrient-depleted region, which is characterized by higher cell death relative to other regions in the colony biofilm (Fig. 2d)[33]. In

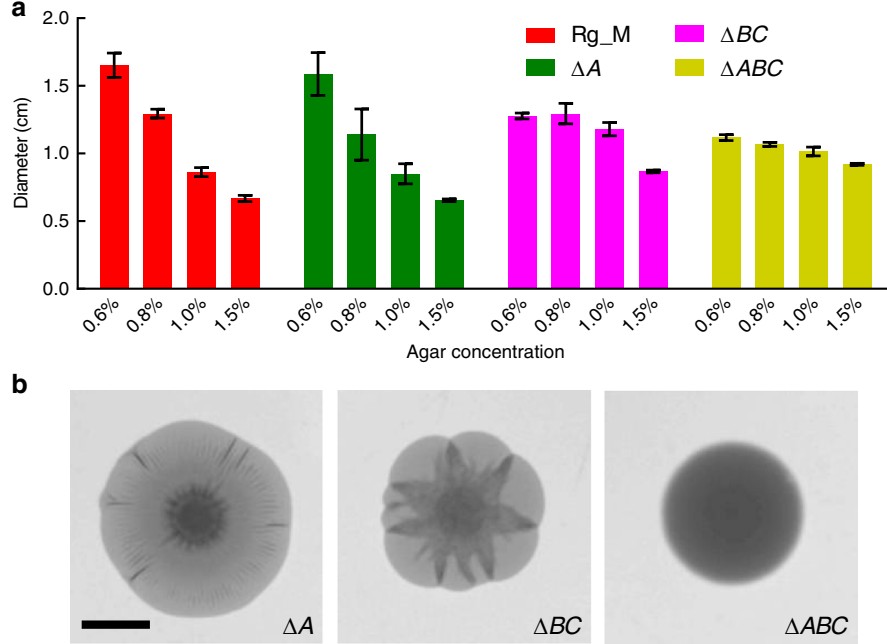

**Fig. 3** Matrix proteins modulate osmotic-pressure-driven expansion of *V. cholerae* colony biofilms. **a** Colony biofilm diameter as a function of agar concentration for the Rg_M (*red*), Δ*rbmA* (denoted Δ*A*, *green*), Δ*bap1*Δ*rbmC* (denoted Δ*BC*, *magenta*), and the Δ*rbmA*Δ*bap1*Δ*rbmC* (denoted Δ*ABC*, *yellow*) strains grown for 2 days on LB medium containing the specified concentrations of agar. **b** Representative images of 2-day-old colony biofilms of the specified mutants on LB medium containing 0.6% agar. *Scale bar*: 0.5 cm. All *error bars* correspond to standard deviations with $n = 4$

regions II and III, the EPS hydrogel network swells and takes up nutrients, which increases cell growth. The colony biofilm first extends in a 3D fashion (region II), but subsequently, transitions to a 2D thin sheet (region III) attached to and expanding on top of the agar. This 3D-to-2D transition is possible only for colony biofilms possessing sufficiently strong internal stiffness and adhesion to the substrate, both of which are provided by the *V. cholerae* matrix. This 2D growth mode contrasts with the 3D droplet growth mode proposed to underpin the expansion of *B. subtilis* biofilms[14]. The characteristic radial ridges in the 2D growth mode, we believe, are formed by localized delamination of the biofilm from the substrate, similar to a mechanical blister[29, 30, 34, 35].

**Osmotic expansion is influenced by matrix proteins.** To provide further support for the model in Fig. 2c, we altered the physical properties of the biofilm matrix to reduce elasticity and/or surface adhesion using mutants in biofilm protein components. The Yildiz group identified three proteins required for the assembly of the *V. cholerae* matrix and proper biofilm formation[36]. The RbmA protein links mother-daughter cells together[37, 38]. The RbmC and Bap1 proteins play partially redundant roles, with RbmC being primarily responsible for interaction with the EPS and Bap1 being primarily responsible for surface adhesion[38–40]. To investigate the contributions of these proteins to osmotic-pressure-driven colony biofilm expansion, we generated Δ*rbmA*, Δ*rbmC*, and Δ*bap1* single, double, and triple mutants in the Rg_M background, and we repeated the colony expansion experiment from Fig. 1a, b. In terms of colony size, the Δ*rbmA* mutant does not differ significantly from the parent (Fig. 3a). By contrast, if both *bap1* and *rbmC* are deleted, in either the Rg_M or the Δ*rbmA* background, the pattern of decreasing colony size with increasing agar concentration is significantly attenuated. We hypothesize that in order for the colony biofilm to expand as a thin sheet, rather than cells simply piling up, strong adhesion to the substratum is required, and this function is

provided by Bap1 and RbmC. Indeed, consistent with the primacy of Bap1 in surface adhesion, the single Δ*bap1* mutant exhibits a more severe phenotype than the single Δ*rbmC* mutant (Supplementary Fig. 3).

The mutant colony biofilm morphologies are also qualitatively different, most notably on the agar plates containing 0.6% agar (Fig. 3b). Specifically, the Δ*rbmA* mutant colony biofilms form much smaller ridges in region III than the Rg_M parent[29, 30], consistent with the reduced mechanical strength of Δ*rbmA* biofilms[31]. The Δ*bap1*Δ*rbmC* double-mutant colony biofilms exhibit a curious star-shaped pattern: each delaminated ridge expands into a wedge shape rather than a straight line, consistent with reduced adhesion of the colony biofilm to the surface. The Δ*rbmA*Δ*bap1*Δ*rbmC* triple-mutant colony biofilm is completely smooth. We propose that, similar to the early stage of *B. subtilis* biofilm development and the mucoid phenotype of *Pseudomonas aeruginosa*[14, 41], in the absence of the matrix protein components, the remaining polysaccharide EPS network is still capable of generating an osmotic pressure contrast, albeit reduced compared to the wild type (Fig. 3a, *yellow bars*). However, the polysaccharide component of the matrix alone does not possess sufficient elasticity or surface adhesion to support 2D expansion. In summary, the protein and polysaccharide components of the extracellular matrix combine to foster the characteristic two-dimensional, osmotic-pressure-driven expansion of *V. cholerae* colony biofilms on nutritious surfaces.

**Osmotic expansion defends biofilms against cheaters.** The enhanced nutrient uptake enabled by osmotic pressure contrast could be considered a mechanism that supplies public goods, benefitting all the cells embedded in the matrix[21, 42]. If so, an interesting question arises concerning whether "cheater" cells— i.e., those which do not produce matrix components—can take advantage of this public good without paying the cost of matrix production[43–45]. To probe this possibility, we used the Δ*vpsL* strain as the putative cheater (Supplementary Fig. 4)[46]. We

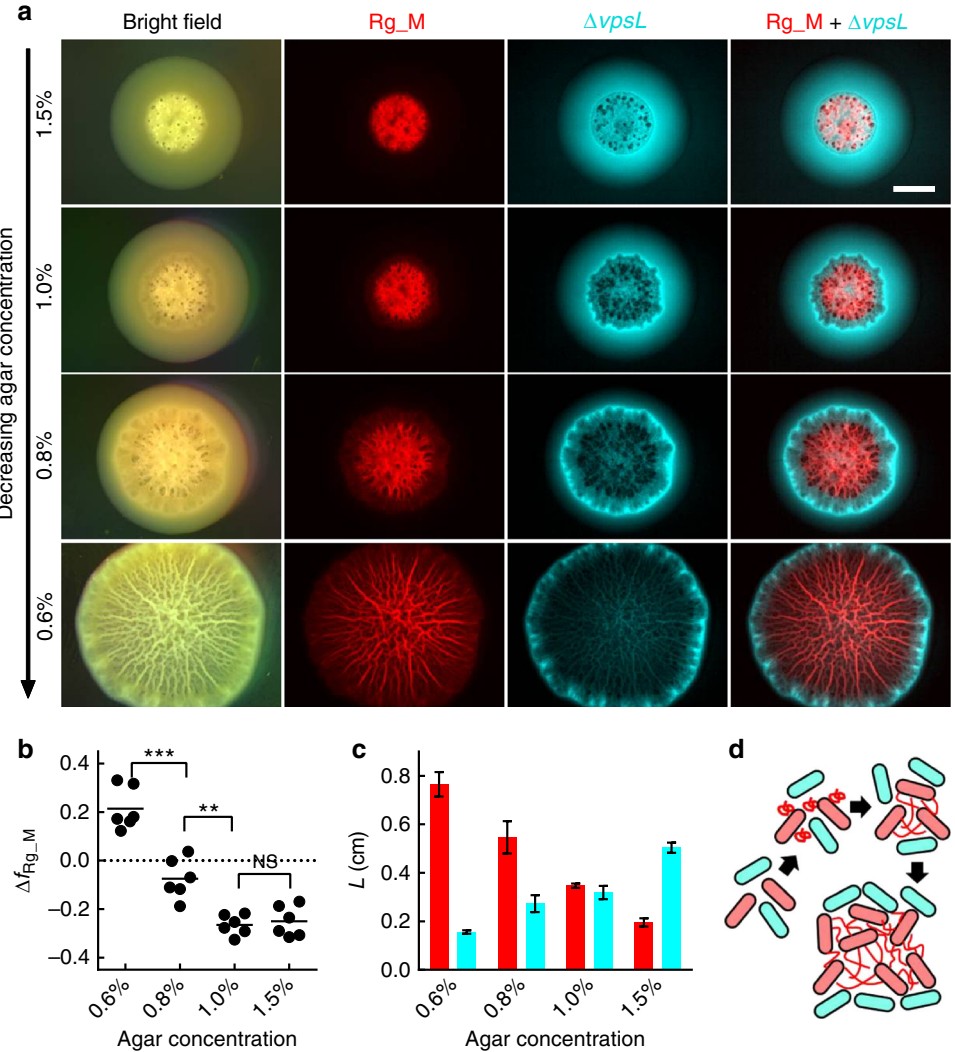

**Fig. 4** Osmotic pressure confers a growth advantage to the *V. cholerae* matrix producer over the non-producer via physical exclusion. **a** Images of 2-day-old colony biofilms formed on LB medium with decreasing agar concentration (*top* to *bottom*) formed from an initial inoculum containing a 1:1 mixture of the Rg_M strain (*red*) and the Δ*vpsL* strain (*cyan*), obtained using stereomicroscopy in bright field mode (*left-most column*), a red filter (*second column*), a cyan filter (*third column*), and overlaid images from the red and cyan channels (*right-most column*). *Scale bar*: 0.3 cm. **b** Change in frequency of the Rg_M strain ($\Delta f_{Rg\_M}$) during a 2-day competition with the Δ*vpsL* strain on agar plates containing the designated concentrations of agar. The initial Rg_M frequency was ~0.5. **c** Quantitation of the sizes of the region $L$ spanned by the Rg_M strain (*red*) and by the Δ*vpsL* strain (*cyan*) as a function of agar concentration. For the Rg_M strain, $L$ corresponds to the radius to which the Rg_M strain extended. For the Δ*vpsL* strain, $L$ corresponds to the width of the annulus in which the Δ*vpsL* strain is present as in **a**. Unpaired *t*-tests with Welch's correction were performed for statistical analyses. NS denotes not significant; **denotes $P$ < 0.01 and ***denotes $P$ < 0.001. *Error bars* correspond to standard deviations with $n = 6$. **d** Schematic representation of the proposed physical exclusion process. *Red* and *cyan cells* denote the matrix producer and non-producer, respectively. *Red wavy lines* depict matrix polymers

co-inoculated the Rg_M strain with the Δ*vpsL* strain at a 1:1 ratio and performed competition experiments on LB medium containing different agar percentages for 2 days at 37 °C (Fig. 4a). On 1.5% agar, the Rg_M strain was over grown by the Δ*vpsL* mutant, restricting the Rg_M strain to the nutrient-depleted region at the center of the colony biofilm. Upon decreasing the agar percentage, the Rg_M strain expanded in size, and, strikingly, physically displaced the Δ*vpsL* strain, relegating it to the periphery of the expanding colony biofilm.

We analyzed the composition of the colony biofilms to quantify the competition results. Increased growth of the Rg_M strain compared to the Δ*vpsL* mutant occurred with decreasing agar percentage (Fig. 4b). Specifically, in competition on 1.5% agar, the absolute frequency of the Rg_M strain decreased by 0.25, whereas on 0.6% agar, its absolute frequency increased by 0.21. Consistent with the cell counts, Fig. 4c shows the quantitation of

the exclusion effect, demonstrating that the sizes of the regions occupied by the Δ*vpsL* cells decrease as the Rg_M colonies increase their osmotic-pressure-driven expansion. Figure 4d shows a schematic representation of the hypothesized mechanism underlying the exclusion process. Following secretion by a producer cell, EPS components bind to the producing cell itself and to nearby cells, which, due to spatial constraints on movement, are most likely to also be EPS producers[43]. The EPS network among producer cells physically swells locally if an osmotic pressure contrast with the environment exists, and in doing so, pushes away the EPS non-producers[47]. We conclude that, despite the possibility of being exploited by the non-producer cells, matrix-producing cells use the EPS network to exclude cheater cells and preferentially consume the nutrient obtained by the osmotic-pressure-driven process to outcompete cheater cells[48].

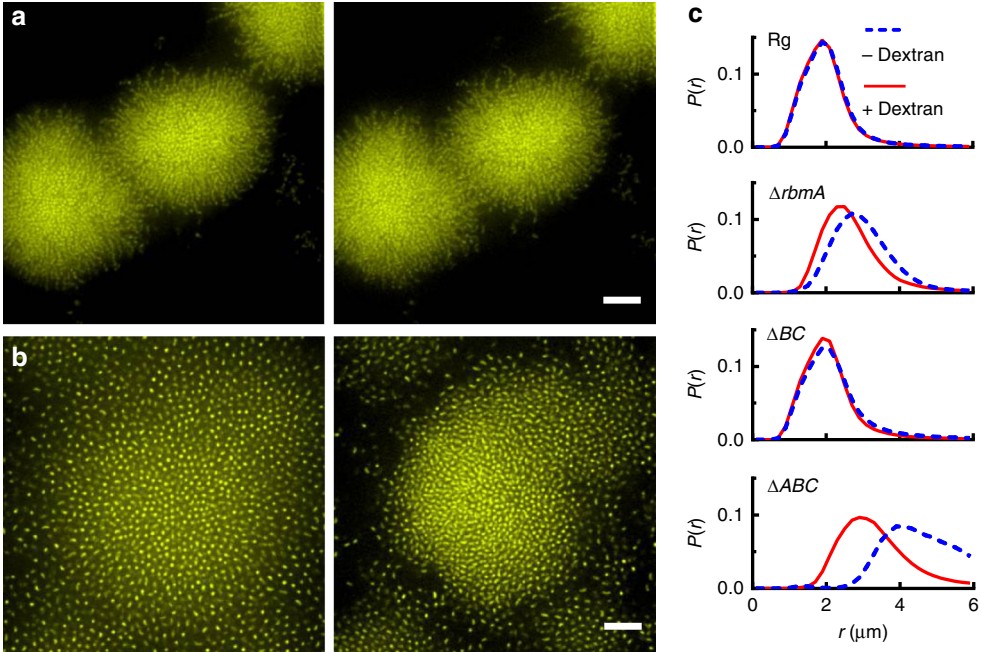

**Fig. 5** Osmotic pressure controls the expansion of submerged *V. cholerae* biofilms. **a**, **b** Cross-sectional confocal images of submerged **a** Rg and **b** Δ*rbmA* biofilms before (*left*) and immediately after (*right*) hyperosmotic shock generated by addition of 15% dextran to the media. Cells constitutively express *mKO* from the chromosome. Images are at 6 and 10 μm above the surface in **a** and **b**, respectively. *Scale bars*: 10 μm. **c** Probability distribution of the cell-to-cell distance *r* in the designated submerged biofilms, before (*dashed blue curve*) and immediately after (*solid red curve*) hyperosmotic shock

**Response of submerged biofilms to osmotic pressure differences.** In addition to colony biofilms on surfaces, *V. cholerae* also frequently exists in submerged biofilms[18, 36]. The osmotic pressure contrast effects we observed above for *V. cholerae* colony biofilms at air-solid or air-liquid interfaces could also be relevant for submerged biofilms. Directional cell proliferation and tight cell packing are the main processes underlying the compact development of 3D submerged *V. cholerae* Rg biofilms[20, 38, 49]. By contrast, the Δ*rbmA* strain forms larger, loosely packed submerged biofilms with increased cell-to-cell distances[20, 38, 49]. We have suggested previously that in the Δ*rbmA* biofilm, increased expansion of the EPS fills the spaces between cells[20]. Our present results lead us to hypothesize that the enhanced polymer expansion in the Δ*rbmA* submerged biofilm is osmotic in origin: the presence of a high local concentration of polymers generates an osmotic pressure differential that draws water into the biofilm, forcing the cells apart. By contrast, polymer expansion in the submerged Rg biofilms is restricted by the rigid scaffold formed by RbmA-mediated cell-to-cell linkages.

To prove that osmotic-pressure-driven polymer expansion controls the Rg and Δ*rbmA* submerged biofilm architectures, we measured their responses to osmotic pressure contrast changes using confocal microscopy[50]. Addition of 15% dextran to the growth medium increases the osmolarity of the medium by ~50 mOsm as measured by vapor pressure osmometry (Supplementary Fig. 1). Such an addition causes a hyperosmotic shock to an existing submerged biofilm. This hyperosmotic shock caused minimal change to the sizes of submerged Rg biofilms (Fig. 5a), whereas the same shock caused submerged Δ*rbmA* biofilms to dramatically shrink (Fig. 5b). Shrinkage must be a physical response of the polymer network to the osmotic pressure change because the experimental time scale (< 5 min) was too short for biological responses such as an alteration in matrix production.

We measured the change in the probability distribution *P*(*r*) of the cell-to-cell distance *r* in the different submerged *V. cholerae* biofilms (Fig. 5c). Rg biofilms have an average *r* around 1.9 μm

that changes minimally (< 2%) upon hyperosmotic shock. The peak *P*(*r*) of the Δ*rbmA* submerged biofilm upon hyperosmotic shock shifts from 3.0 to 2.6 μm, corresponding to a volume decrease of ~32% (Fig. 5c and Supplementary Fig. 5). The behavior of the Δ*bap1*Δ*rbmC* submerged biofilm is similar to that of the Rg strain: the peak position (2.0 μm) changes minimally upon hyperosmotic shock (< 5%). The effect of hyperosmotic shock on the Δ*rbmA*Δ*bap1*Δ*rbmC* submerged biofilm is the largest with a peak *r* of 4.4 μm that shifts to 3.2 μm corresponding to a 60% decrease in volume (Fig. 5c and Supplementary Fig. 5).

We interpret the above results starting with the case of the Δ*rbmA*Δ*bap1*Δ*rbmC* triple mutant, which contains the least crosslinked EPS network. Upon hyperosmotic shock, the increasing osmotic pressure in the external environment drives water out of the biofilm, causing the matrix to shrink. With respect to the Δ*rbmA* submerged biofilm, in this case, RbmC and Bap1 are present and are known to interact with the polysaccharide component of the matrix[38]. We suggest that these interactions can be conceptualized as providing crosslinks within the EPS network and, in so doing, they limit the swelling of the EPS network during biofilm formation and thus attenuate its response to hyperosmotic shock. With respect to the Δ*bap1*Δ*rbmC* mutant and the Rg parent strain, in these cases, RbmA is present and it connects the cells to one another[38]. We suggest that RbmA creates a dense scaffold of cells to which the EPS adheres. This stiff cell-based scaffold strongly restricts the expansion of the EPS in the Rg and Δ*bap1*Δ*rbmC* strains during submerged biofilm growth, and consequently, limits shrinkage of the EPS upon hyperosmotic shock. The presence/absence of this cell-based scaffold underlies the differences in the submerged biofilm architectures[20], with the Rg architecture driven primarily by extension of the cellular scaffold due to directional cell division and the Δ*rbmA* architecture driven primarily by osmotic-pressure-mediated enlargement of the EPS network.

We can estimate the osmotic pressure Π acting on the submerged biofilms by calculating the total polymer volume

fraction $\phi$ in the biofilm, which is roughly 0.04 for the Rg strain (Supplementary Discussion). According to a scaling law for a semi-dilute polymer solution of ideal chains[51, 52], we estimate $\Pi_{EPS} \sim kT/b^3 \times \phi^3$, in which $kT$ is the thermal energy, $b$ is roughly the linear size of a sugar monomer (~0.5 nm), and the $\phi^3$ dependence arises from polymer-polymer excluded volume. This treatment yields a value $\Pi_{EPS} \sim 2$ kPa, which falls well within the range measured for biofilm elastic moduli (10–10,000 Pa)[53, 54]. The strong (cubic) dependence of $\Pi_{EPS}$ on $\phi$ means that the elastic modulus of the $\Delta rbmA$ strain, for which $\phi \sim 0.01$, will be ~64 times lower than that of the Rg strain, which explains the much reduced mechanical strength of the $\Delta rbmA$ biofilm[20, 31].

**Osmotic pressure modulates biofilm resistance to invaders.** In nature, biofilms are challenged by invader cells that can take advantage of public goods produced by the cells residing in the existing biofilm community[55, 56]. Invader cells can be distinguished from cheater cells because invaders encounter surfaces occupied by existing biofilms. A key determinant underlying resistance to invasion is biofilm compactness[55]. In the context of our current work, the reduction in cell-to-cell distance that occurs in the $\Delta rbmA$ submerged biofilms upon hyperosmotic shock suggested to us that vulnerability to invader cells could also be controlled by osmotic pressure. To test this hypothesis, we performed an invasion assay using motile $\Delta vpsL$ cells as the invaders and the $\Delta rbmA$ strain or the Rg strain as the resident submerged biofilm. Consistent with previous results[55], invader cells could not penetrate the submerged Rg biofilms but could invade the submerged $\Delta rbmA$ biofilms (Fig. 6a, b). However, in the presence of 15% dextran, the $\Delta rbmA$ resident biofilms became compact and, importantly, resistant to invasion (Fig. 6c). To quantify these results, we analyzed the distribution of invader cells in the resident biofilms as a function of penetration depth (Fig. 6d). The Rg curve has a sharp decay: the number of invader cells declines by more than a factor of three by a depth of 3 μm into the submerged biofilm. By contrast, the $\Delta rbmA$ curve shows a much slower decay. Indeed, invader cells are present deep within the biofilm (>30 μm). Upon the addition of 15% dextran, however, the $\Delta rbmA$ invasion curve shows a sharper decay, approaching that of the Rg submerged biofilm. This observation is consistent with the data in Fig. 5c: the cell-to-cell distance decreases in the submerged $\Delta rbmA$ biofilm in the presence of dextran, approaching the cell-to-cell distance in the Rg submerged biofilm. We note that the increased viscosity of the osmolyte solution could further reduce invasion by slowing the swimming speed of the invaders[55]. However, our control experiment shows that the presence of dextran does not affect invasion of the Rg-resident biofilms under our experimental conditions, suggesting that swimming speed is not the limiting factor in invasion (Supplementary Fig. 6).

**Discussion**

We have demonstrated here the importance of osmotic pressure generated by the hydrogel-like *V. cholerae* EPS matrix to the growth and surface coverage of *V. cholerae* biofilms residing at air-solid interfaces and submerged in liquids. The physical principles underlying the behavior of this biological hydrogel should be applicable to other biofilm matrices, although the exact expansion/shrinkage characteristics due to changes in osmotic pressure will vary depending on the specific biochemistries of the different matrix polysaccharides and the functional roles of the participating proteins. The success of cheater and invader cells is strongly affected by osmotic pressure, which could provide an explanation for reports that submerged biofilms can exclude matrix non-producers[55, 57, 58]. Our results, furthermore,

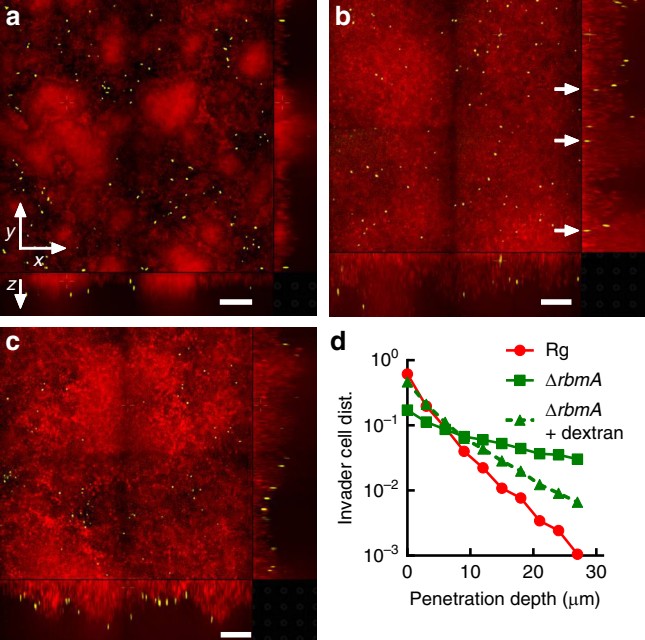

**Fig. 6** Osmotic-pressure-driven swelling modulates *V. cholerae* submerged biofilm susceptibility to invasion. **a–c** Representative cross-sectional confocal images of resident biofilms carrying *mKate2* (*red*) invaded for 10 min by *ΔvpsL* strain carrying *mKO* (*yellow*). The resident biofilm strain is Rg in **a** and *ΔrbmA* in **b** and **c**. Invasions were performed in M9 medium without dextran in **a** and **b** and with 15% dextran in **c**. Shown are the *xy* cross-sectional views in the *upper left* panels and the two orthogonal views in the *bottom* and *right* panels. *White arrows* in **b** are guides to show invader cells that have successfully penetrated the resident *ΔrbmA* submerged biofilm. *Scale bars*: 30 μm. **d** Distributions (denoted dist.) of invader cells as a function of penetration depth into the resident Rg (*red circles*), *ΔrbmA* (*green squares*), and *ΔrbmA* submerged biofilms in the presence of 15% dextran (*green triangles*). Distributions are normalized by the total number of invader cells in each case

underscore the importance of understanding the physical forces that together with biochemical and genetic features drive biofilm ecology. The physical responses of the biofilm matrix to osmotic pressure offer possibilities to use external osmolytes to control the accessibility of small molecules such as antibiotics and signaling compounds[59] to biofilm-dwelling cells. Such manipulation could be used to remove harmful biofilms or to enhance the robustness of beneficial biofilms by osmotically weakening or strengthening the biofilm structures, respectively.

With respect to the relevance of our findings to the natural ecology of *V. cholerae*, this bacterium resides in fluctuating environments that undergo extreme changes in osmotic pressure[60, 61]. Indeed, *V. cholerae* cycles between the ocean and fresh water aquatic environments and it resides in the human intestine during infection[62]. *V. cholerae* forms robust biofilms in each of these niches[23, 63, 64]. We suggest that a scaffold is formed by RbmA-mediated interconnections among the cells, and this structure plays an overarching role in restricting the osmotic expansion of the underlying polysaccharide network. Recently, a new intracellular niche was discovered for *V. cholerae* in the marine amoeba *Acanthamoeba castellanni*[65]. Interestingly, *V. cholerae* cells survive and replicate in this host's contractile vacuole, a dynamic organelle involved in the osmoregulation of the amoeba. Following intracellular replication, wild-type *V. cholerae*, but not EPS mutants, can destroy this organ and disperse from the host. We speculate that the capacity of the wild type to successfully disseminate from the vacuole could be related

to the osmotic pressure response of the biofilm matrix shown here. Similar osmotic effects could also influence *V. cholerae* colonization of the mucus layer in the human intestine, since mucus is a bio-hydrogel that expands and contracts in response to polymer concentrations in the gut[50].

## Methods

**Strains and media**. All *V. cholerae* strains used in this study are derivatives of the wild-type *V. cholerae* O1 biovar El Tor strain C6706, harboring a missense mutation in the *vpvC* gene (VpvC W240R)[19]. Additional mutations were engineered into this strain using *Escherichia coli* S17 λ*pir* carrying pKAS32[66]. All strains were grown in LB medium at 37 °C with shaking. When designated, fresh LB medium solidified with different percentages of agar was used. Submerged biofilms were grown in M9 minimal medium, supplemented with 2 mM MgSO$_4$, 100 μM CaCl$_2$, and 0.5% glucose as the carbon source. To generate hyperosmotic shock to submerged biofilms, 15% dextran (1500–2800 kDa, Sigma-Aldrich) solutions were prepared by adding dextran powder to M9 medium or LB medium followed by shaking at 30 °C for at least 3 h. A detailed strain list is provided in Supplementary Table 1.

**Statistics**. Throughout the text, error bars correspond to standard deviations of the means. As per convention in the field, standard *t*-tests were used to compare treatment groups and are indicated in each figure legend. Tests were always two-tailed, and were paired or un-paired as demanded by the details of the experimental design.

**Confocal microscopy**. Images were acquired with a Yokogawa CSU-X1 confocal spinning disk unit mounted on a Nikon Ti-E inverted microscope, using a ×60 water objective with a numerical aperture of 1.2 and an Andor iXon 897 EMCCD camera. For the hyperosmotic shock experiment, a 543 nm laser (OEM DPSS) was used to excite cells expressing *mKO*, and a ×1.5 post-magnification lens was used to obtain sufficient magnification for automated cell segmentation. The magnification was 166 nm/pixel in the *xy* plane with a 200 nm step size in the *z* direction. The point spread function of the system was measured under identical conditions (with 50 nm *z*-step size) using 200 nm fluorescent polystyrene beads (ex540/em561 Life Technologies). For the invasion experiment, two diode lasers (543 and 592 nm) were used sequentially along with customized dichroic filters (Chroma) for spectral separation. For live-dead staining, 488 and 592 nm lasers were used combined with the appropriate filter sets. All experimental images in this work are raw images rendered by Nikon Element software.

**Colony biofilm growth on agar plates**. *V. cholerae* strains were streaked on LB plates containing 1.5% agar and grown at 30 or 37 °C overnight. Individual colonies were selected and inoculated into 3 mL of LB liquid medium containing glass beads, and the cultures were grown with shaking at 37 °C to mid-exponential phase (5–6 h). Subsequently, the cells in the cultures were mixed by vortex, OD$_{600}$ was measured, and the cultures were back diluted to an OD$_{600}$ of 0.5. An aliquot of 1 μL of this inoculum was spotted onto pre-warmed agar plates solidified with different percentages of agar. The plates were inverted and incubated at 37 °C for 2 days. Four colonies were grown per agar plate. After 2 days of growth at 37 °C, the entire plate was imaged with an ImageQuant LAS-4000 gel imager (GE Healthcare) in transmission mode. Nikon Element software was used to quantify the diameters of the colony biofilms. When semipermeable membranes were included, the procedure was identical except the inocula were applied to EMD Millipore, VSWP02500 semipermeable membranes that had been placed on top of the agar. Also, growth was extended to 4 days, and imaging was performed in the reflection mode since the semipermeable membranes are not transparent. Two technical replicates were performed for each biological replicate, and four biological replicates were performed. In the experiments in Fig. 1e, agar plates were transferred to 4 °C (denoted Control) and the semipermeable membranes harboring the colony biofilms were floated in 100 mm petri dishes containing 25 mL of liquid LB overnight at 4 or 37 °C. The semipermeable membranes were subsequently transferred to a plastic tray and imaged with the gel imager. One technical replicate was performed for each biological replicate and four biological replicates were performed. The contact angle measurements were performed using a home-built set-up. Side views of water droplets (1 μL) were recorded with a Nikon camera (D90) equipped with a macro lens. Images were analyzed using the Droplet_Analysis plugin in ImageJ.

**Submerged biofilm growth**. *V. cholerae* strains were grown overnight at 37 °C in liquid LB medium with shaking, back-diluted 30-fold, and grown for an additional 2 h with shaking in M9 medium until early exponential phase (OD$_{600}$ = 0.1–0.2). To form isolated submerged biofilm clusters, these regrown cultures were diluted to OD$_{600}$ = 0.001 and 100 μL of the diluted cultures were added to wells of 96-well plates with #1.5 coverslip well bottoms (MatTek). The cells were allowed to attach for 10 min, after which the wells were washed twice with fresh M9 medium, and, subsequently, 100 μL of fresh M9 medium was added. The inoculated wells were

incubated at 30 °C for 16–18 h. The resulting submerged biofilms were washed twice and, subsequently, 100 μL of fresh M9 medium lacking glucose was added to terminate cell growth for 2–3 h prior to imaging. To form resident biofilms for invasion experiments, the inoculating OD$_{600}$ was adjusted to 0.1, the initial inoculation time was extended to 1 h, and the submerged biofilm growth time was extended to 20 h so that confluent biofilms could form.

**Hyperosmotic shock experiment**. Submerged biofilms were grown as described above, and imaged with confocal microscopy. Immediately after imaging, the medium was replaced with fresh M9 medium lacking glucose but containing 15% dextran, and the wells were imaged again. Whenever possible, the same biofilms were imaged prior to and after hyperosmotic shock. As noted in the legend of Supplementary Fig. 5, this was not always possible with the Δ*bap1*Δ*rbmC* and Δ*rbmA*Δ*bap1*Δ*rbmC* biofilms because they adhere only poorly to the substratum. In these cases, different biofilms were imaged pre- and post-hyperosmotic shock.

Automatic cell segmentation steps were performed as described[20]. After segmentation, the position of each cell was recorded. 3D Delaunay triangulation[67] was performed to identify neighboring cell pairs, with a threshold distance of 6 μm. Due to surface attachment, cells close to the substratum pack more densely than those in the bulk, and this effect is most prominent in Δ*rbmA* biofilms. To avoid this substratum effect, we only included cells more than 6 μm away from the surface in our analyses; the exact threshold, however, does not significantly affect our conclusions. For each biofilm cluster, a Gaussian fit to the probability distribution of the cell-to-cell distance *r* was applied to obtain the average *r*. The fold change in volume was obtained from the cube of the ratio of *r* before and after hyperosmotic shock. Three clusters were analyzed for each biological replicate, and four biological replicates were performed for each strain. The data presented in Fig. 5c were obtained by combining all neighboring cell pairs in each biofilm from different experiments.

**Swelling of an osmolyte solution on an agar plate**. Semipermeable membranes (EMD Millipore, VSWP01300) were placed on LB plates containing different agar percentages for at least 30 min to allow them to equilibrate to room temperature. Each plate was placed on a balance, and a 20 μL aliquot of dextran solution in LB was applied to the top of the semipermeable membrane, and its weight recorded. Swelling was allowed to proceed for the designated times at room temperature, and subsequently, the membrane along with the solution was transferred onto a balance. The total weight of the droplet plus the membrane was measured, and the weight of the membrane was calibrated separately. Using this method, the instantaneous weights of the droplets were obtained. Three replicates were performed for each condition and for each time point, and each replicate contained two technical replicates.

**Quantification of cell death in the colony biofilms**. *V. cholerae* colony biofilms were grown for 2 days at 37 °C as described above. Different regions of the colonies were isolated using a pipette tip and the cells were resuspended in M9 medium lacking a carbon source but containing a 1:1000 dilution of the LIVE/DEAD *Bac*Light bacterial viability kit components (Thermo Fisher). The suspension was mixed using small glass beads (Acid-washed, 425–500 μm, Sigma) at 37 °C for 5 min, and subsequently, 5 μL of the suspension was placed between a #1.5 coverslip and an agarose gel pad, and the cells were imaged by confocal microscopy with a ×100 oil objective (numerical aperture = 1.4). The number of cells in ten 100 × 100 μm sections of each region from two colonies was quantified for each biological replicate. Four biological replicates were performed. We note that it was not technically possible to isolate region II from region III because they are continuous. Automatic cell finding codes were written in Matlab to track the number of live (green) and dead (red) cells in these regions and their frequencies were subsequently calculated.

**Competition experiments between matrix-producing cells and matrix-non-producing cheater cells**. Mid-exponential phase cultures of *V. cholerae* strains were prepared as described above. Cultures were transferred to Eppendorf tubes containing glass beads (Acid-washed, 425–500 μm, Sigma), and vigorously mixed by vortex. These steps were necessary to disassemble cell clusters that had formed in the liquid cultures enabling accurate OD$_{600}$ measurement and equalization. An inoculum containing a total number of cells equal to OD$_{600}$ = 0.5 was prepared with 1:1 mixture of the two competing strains. An aliquot of 1 μL of this inoculum was spotted onto pre-warmed agar plates solidified with different percentages of agar. The initial inoculum was temporarily frozen after mixing with 80% glycerol, and the exact frequency $f_{0,Rg\_M}$ in the inoculum was measured at a later point. After 2 days of colony growth at 37 °C, the entire colony was carefully harvested with a pipette tip and suspended in 1 mL of M9 medium lacking a carbon source, mixed using glass beads for 30 min at 37 °C, serially diluted in M9 medium lacking a carbon source, and subsequently, plated on LB plates containing 1.5% agar. Colonies were allowed to grow overnight at 30 or 37 °C and subsequently imaged with a gel imager with different filter sets (Cy2 and Cy5) to differentiate cells bearing *mKate2* from those carrying *mTPF1*. Colonies of each strain were counted using Nikon Element software, and the final frequency $f_{Rg\_M}$ calculated. The change in frequency $\Delta f_{Rg\_M}$ was defined as $f_{Rg\_M} - f_{0,Rg\_M}$.

In separate experiments, the 2-day-old colonies containing the mixtures of competing strains were directly imaged with the gel imager and the appropriate filter sets (Cy2 and Cy5). The sizes of the regions occupied by the different strains were quantified using Nikon Element software. All competition experiments were repeated three times using Rg_M cells harboring *mKate2* and Δ*vpsL* cells harboring *mTFP1*. The experiment was repeated three times with the fluorescent markers exchanged. Each biological replicate in the competition experiment contained two technical replicates. Images in Fig. 4a were obtained using a Leica stereoscope with CFP and mCherry filter sets or with no filter (reflection mode).

**Invasion of resident biofilms**. Confluent resident biofilms were grown according to the above procedure. Invader cells were grown overnight at 37 °C in liquid LB medium with shaking, back-diluted 30-fold, and grown for an additional 3 h with shaking in LB medium until mid-exponential phase ($OD_{600} = 0.5$–1.0). Cultures of known $OD_{600}$ were transferred to Eppendorf tubes, pelleted, and washed once with PBS. Fresh M9 medium (no glucose) with or without 15% dextran was added to make the final $OD_{600}$ of 1.0. The Eppendorf tubes were subjected to vigorous vortex for 10 min to resuspend the cells. Invader cells (100 μL) were added to the wells containing the resident biofilms. The invasion process was allowed to proceed for 10 min, and subsequently, the cell suspension was taken out and the resident biofilms were washed three times prior to imaging. Six different locations each with a total size of $250 \times 250$ μm with a $z$-interval of 3 μm were imaged in each biological replicate. The total imaging height was 60 μm if Δ*rbmA* cells were the resident biofilm or 36 μm if the Rg cells were the resident biofilm. Each invasion condition was repeated twice with resident biofilm cells harboring *mKate2* and invading cells harboring *mKO*. The experiment was performed twice with the fluorescent markers reversed. Invader cell numbers were analyzed with custom-written Matlab codes.

**Data availability**. The data supporting the findings of this study are available from the corresponding authors on request. Custom-written Matlab scripts used in this study are available at https://github.com/yanjing32/Osmotic-Pressure-Project.

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

## Acknowledgements

This work was supported by the Alexander von Humboldt Foundation (C.D.N.), Howard Hughes Medical Institute (B.L.B.), National Science Foundation Grants MCB-0948112 (B.L.B) and MCB-1344191 and the Eric and Wendy Schmidt Transformative Technology Fund (to N.S.W., B.L.B., and H.A.S.), NIH Grant 2R37GM065859 (B.L.B.), and the Max Planck Society-Alexander von Humboldt Foundation (B.L.B.). J.Y. holds a Career Award at the Scientific Interface from the Burroughs Wellcome Fund. We thank Dr. Thomas Bartlett, Dr. Benjamin Bratton, Ms. Ying Liu, Dr. Sheng Mao, and Dr. Zhong Zheng for helpful discussions and Dr. Emily Zytkiewicz and Professor M. Thomas Record, Jr for help with osmotic pressure measurements.

## Author contributions

J.Y. and B.L.B. initiated this work. J.Y. performed the experiments. J.Y., C.D.N., H.A.S., N.S.W., and B.L.B. analyzed the data. All authors contributed to writing the paper.

## Additional information

**Competing interests:** The authors declare no competing financial interests.

