## [Peer Review File · Nature Communications]

Reviewers' comments:

Reviewer #1 (Remarks to the Author):

Review of "Extracellular-matrix-mediated osmotic pressure drives *Vibrio cholerae* biofilm expansion and cheater exclusion" by Yan et al.

Yan et al conduct a series of experiments on colony as well as submerged biofilms made by *Vibrio cholerae* and demonstrate the critical role of osmotically driven expansion for biofilm growth; morphology; nutrient uptake and resistance to cheaters as well as invaders. They first prove that osmotically driven expansion occurs for this system, as previously proposed for *Bacillus subtilis*. They show that swelling depends on the environmental osmotic pressure; they then corroborate the picture by a bio-mimetic experiment with swelling and shrinking drops of dextran solution in contact with the same agar substrates; finally they grow biofilms on semipermeable membranes to prove that an influx of water occurs when the surrounding osmotic pressure is suddenly decreased. EPS- mutant does not show any osmotically driven swelling.

They then provide physical arguments to parse the role of (i) matrix cross linking, (ii) surface adhesion and (iii) cell-to-cell connection to explain the observed morphology. The wild type transitions from a 3D morphology in the center to a 2D sheet, that develop features due to its elastic nature and adhesion with substrate. This physical picture is validated by an analysis of single, double and triple knock outs of three major proteins that provide functions (i),(ii),(iii). The experiments on agar of different concentration confirm that matrix cross linking and surface adhesion are essential to provide osmotic swelling, whereas cell-to-cell links are unimportant for osmotic spreading. The same physical picture is consistent with their observation that at small enough external osmotic pressure, non-matrix producers are physically excluded from the biofilm when co-inoculated with the wild type.

The same proteins play a major role in the response of submerged biofilms against invading cells. Here, mother-to-daughter links become crucial to prevent over-swelling and invasion.

I enjoyed reading this manuscript and I think it will provide an interesting contribution to the literature. The physical arguments are convincing they are backed up with both scaling analysis and information from the literature. The comparison with mutant strains is solid, and it contributes to parse the apparent complexity of biofilm biomechanics, establishing a clear picture that will be useful for fundamental as well as applied research. I recommend publication of this manuscript in Nature Communication after minor revisions.

Comments:

(1) When biofilms are grown on the semipermeable membrane, both wild type and EPS- slow down. The authors suggest that this results from a slower nutrient uptake due to the presence of the membrane. The argument for the wild type is that osmotically driven flow slows down when crossing the membrane, which is a sensible argument assuming that the pore size of the membrane are smaller than the mesh size of the agar (is there any evidence in this regard? Could the authors mention this underlying assumption?). Instead, the EPS- relies on diffusion for nutrient uptake, and I find it surprising that a semipermeable membrane would slow down diffusion, could the authors elaborate ?

An alternative explanation is that biofilm expansion on the membrane is slowed down by an increase in friction due to different surface properties of agar vs membrane. Could the authors simply rule this out by looking at the contact angles?

(2) The results parsing growth from water influx for the wt are interesting and convincing (fig 1e). the EPS- shows no growth whatsoever. The absence of water inflow is consistent with the idea that no osmotic pressure is exerted by this strain, hence the biofilm is a close packing of cells, both on agar and on liquid no matter how little the external osmotic pressure is. But why don't cells grow? At least the cells in contact with the fresh medium should not be starving.

(3) The results of physical exclusion of the cheaters in Figure 3 are striking. On 0.6%, the EPS-

clearly decreases in frequency relative to the wild type. However, it is confined in a nutrient rich region, so their absolute growth rate should not be smaller than that on 1.5%. Presumably the colony forming units assay provides approximate information on total cell counts if dilutions are accounted for, could the authors quantify total cell counts to elucidate this point?

Typos and the like:

- (1) Data for Rg_M and DeltaBC in Supplementary Fig 3 are different from those in Figure 2, is this a different replica of the same experiments?
- (2) Line 459 where -> were
- (3) supplementary figure 2: using the same color code for Region I in panel b and c would help visually

Reviewer #2 (Remarks to the Author):

The authors present work describing *Vibrio cholerae* colony biofilm formation and the connection between spreading of colonies on agar plates with osmotic pressure differences and the influence on nutrient uptake. The results are clear and the manuscript is easy to read. The motivation for the work and the findings extend from work by Brenner and coworkers on *Bacillus subtilis* (PNAS, 2012). The work presented in this submitted manuscript is solid and valuable for this understudied area of biofilm biology. It provides specific data to support important fundamental concepts and the figures are well designed and illustrate the science. Importantly, the authors include competition experiments wherein they demonstrate that matrix-producing cells outcompete non-matrix producers. Thus, the authors provide comprehensive data documenting the results of matrix production on spreading in *V. cholerae*, an organism that is studied widely by many laboratories. The impact of the work is very high and likely to be of broad interest.

The manuscript would be improved if the authors were clearer on the specific previous findings and hypotheses presented in the Brenner paper and those that are new here. The manuscript does briefly capture the overall nature of the 2012 paper but without any specifics and this gives the reader the sense that doing so could diminish the novelty of this manuscript. To the contrary, the clear statements of previous results and hypotheses would strengthen this manuscript and the generality of the findings and the growing sets of questions. The following specific comments involve these minor issues.

1. At the top of page 3, the authors state: "In biofilms, the high local concentration of polymer molecules surrounding the cells necessarily produces an osmotic pressure difference between the matrix and the external environment. It is not clear if or how such osmotic pressure gradients influence the mechanical properties of biofilms or the growth and fitness of the bacteria residing in them." The first statement could be supported with a reference (Brenner or earlier reference). The osmotic pressure difference also depends, in principle, on the exact external environment (aqueous vs hypersaline vs sludge environments). Can the authors comment on this?

2. In the following sentence, the authors accurately state: "Experimental and theoretical work by Seminara et al. analyzing *Bacillus subtilis* colonies on agar plates suggested a crucial role for EPS-generated osmotic pressure differences in facilitating nutrient uptake¹⁴." However, the authors later state that: "We hypothesize that, relative to passive diffusion, an osmotic pressure-driven transport process could accelerate nutrient uptake from resources located at a distance from the biofilm cells, because transport times scale linearly with distance in osmotic-driven processes but quadratically in diffusion-limited processes²¹." This notion of an osmotic pressure-driven transport process, however, is exactly what was concluded in the Brenner paper, wherein they discussed a result that "strongly suggests that the energetic investment implied in the production of EPS is rewarded by the consequent increase in nutrient uptake and results in a net fitness increase for the colony." Their abstract even ended with the statement "the implications of this osmotically

driven type of surface motility for nutrient uptake that may elucidate the reduced fitness of the matrix-deficient mutant strains." The authors of this work on *V. cholerae* should specifically describe the major hypotheses and results that were published in the 2012 paper.

Response to Reviews:

We are pleased that both reviewers found our central findings to be of broad interest and general importance. The reviewers kindly provided us suggestions to clarify some of our findings and to better emphasize the significance of the work in the context of the existing literature. We have taken these comments to heart and revised the manuscript exactly as requested. Specifically, we made all of the suggested textual changes and we now provide additional experiments and data according to the reviewers' requests. A point-by-point list of our responses to the reviewers' comments is provided below. The reviewers' comments are in black text and our responses are in blue.

Reviewer #1:

Yan et al conduct a series of experiments on colony as well as submerged biofilms made by *Vibrio colerae* and demonstrate the critical role of osmotically driven expansion for biofilm growth; morphology; nutrient uptake and resistance to cheaters as well as invaders. They first prove that osmotically driven expansion occurs for this system, as previously proposed for *Bacillus subtilis*. They show that swelling depends on the environmental osmotic pressure; they then corroborate the picture by a bio-mimetic experiment with swelling and shrinking drops of dextran solution in contact with the same agar substrates; finally they grow biofilms on semipermeable membranes to prove that an influx of water occurs when the surrounding osmotic pressure is suddenly decreased. EPS- mutant does not show any osmotically driven swelling.

They then provide physical arguments to parse the role of (i) matrix cross linking, (ii) surface adhesion and (iii) cell-to-cell connection to explain the observed morphology. The wild type transitions from a 3D morphology in the center to a 2D sheet, that develop features due to its elastic nature and adhesion with substrate. This physical picture is validated by an analysis of single, double and triple knock outs of three major proteins that provide functions (i),(ii),(iii). The experiments on agar of different concentration confirm that matrix cross linking and surface adhesion are essential to provide osmotic swelling, whereas cell-to-cell links are unimportant for osmotic spreading. The same physical picture is consistent with their observation that at small enough external osmotic pressure, non-matrix producers are physically excluded from the biofilm when co-inoculated with the wild type.

The same proteins play a major role in the response of submerged biofilms against invading cells. Here, mother-to-daughter links become crucial to prevent over-swelling and invasion.

I enjoyed reading this manuscript and I think it will provide an interesting contribution to the literature. The physical arguments are convincing they are backed up with both scaling analysis and information from the literature. The comparison with mutant strains is solid, and it contributes to parse the apparent complexity of biofilm biomechanics, establishing a clear picture that will be useful for fundamental as well as applied research. I recommend publication of this manuscript in Nature Communication after minor revisions.

We thank the reviewer for this supportive overview and assessment of our paper.

Specific comments:

(1) When biofilms are grown on the semipermeable membrane, both wild type and EPS- slow down. The authors suggest that this results from a slower nutrient uptake due to the presence of the membrane. The argument for the wild type is that osmotically driven flow slows down when crossing the membrane, which is a sensible argument assuming that the pore size of the membrane are smaller than the mesh size of the agar (is there any evidence in this regard? Could the authors mention this underlying assumption?). Instead, the EPS-

relies on diffusion for nutrient uptake, and I find it surprising that a semipermeable membrane would slow down diffusion, could the authors elaborate ?

An alternative explanation is that biofilm expansion on the membrane is slowed down by an increase in friction due to different surface properties of agar vs membrane. Could the authors simply rule this out by looking at the contact angles?

We thank the reviewer for the insightful comments on this issue. This is when peer-review works at its best! Based on new experiments, we believe that the reviewer has the correct interpretation. To distinguish between the two possibilities, as suggested, we performed the contact angle measurement. The data are provided in the new SI Fig. 2. Indeed, the contact angle on the semipermeable membrane ($\sim 53^\circ$) is much larger than that on the bare agar surface ($\sim 14^\circ$), independent of the agar concentration. This result means that for the EPS⁻ mutant colony, which exists as a 3D droplet, the contact area with the substrate will be smaller if grown on the semipermeable membrane than on bare agar, leading to reduced nutrient flux. The higher frictional forces on the semipermeable membrane could also mechanically impede the expansion of the EPS⁻ colony. We agree with the reviewer that this scenario provides a superior explanation for the more retarded growth of the EPS⁻ colony on the semipermeable membrane compared to the EPS⁺ strain.

To further prove the reviewer's hypothesis that the difference in contact angle/friction of agar versus the semipermeable membrane causes retardation of colony growth in EPS⁻ colonies, we performed another set of experiments. We systematically increased the thickness of the semipermeable membrane on which the colony was growing, by stacking multiple membranes between the colony and the agar. We reasoned that, if the surface property of the membrane plays a more important role in slowing colony expansion than does reduced nutrient transport, the colony size should be insensitive to the total membrane thickness. Indeed, this is the case. Based on this new insight, we have rewritten the entire text corresponding to the slowed growth of the EPS⁻ strain on the semipermeable membrane. We note that the internal comparison between the colony biofilm sizes of EPS⁺ strains grown on semipermeable membranes atop agar of different concentrations remains valid and proves the role of the osmotic differential.

(2) The results parsing growth from water influx for the wt are interesting and convincing (fig 1e). the EPS- shows no growth whatsoever. The absence of water inflow is consistent with the idea that no osmotic pressure is exerted by this strain, hence the biofilm is a close packing of cells, both on agar and on liquid no matter how little the external osmotic pressure is. But why don't cells grow? At least the cells in contact with the fresh medium should not be starving.

We suspect that, in the case of the EPS⁻ cells, they are growing, albeit very slowly, resulting in only a minor increase in colony size during the experiment and that change in size is not statistically significant/meaningful. As mentioned in the main text, the EPS⁻ cells rely exclusively on diffusion to take up nutrients, which is a much slower process than is osmotic pressure driven nutrient uptake in the EPS⁺ cells. The spreading of the EPS⁻ colony is further hampered by the high contact angle of the semipermeable membrane, as suggested by the reviewer and other literature (arXiv:1612.05450, 2016). The EPS⁺ biofilms are less affected by the increase in contact angle, as they still expand as a thin elastic sheet on the semipermeable membrane (rather than as a liquid drop).

(3) The results of physical exclusion of the cheaters in Figure 3 are striking. On 0.6%, the EPS- clearly decreases in frequency relative to the wild type. However, it is confined in a nutrient rich region, so their absolute growth rate should not be smaller than that on 1.5%. Presumably the colony forming units assay provides approximate information on total cell counts if dilutions are accounted for, could the authors quantify total cell counts to elucidate this point?

We thank reviewer for raising this interesting point. The data in SI Fig. 5 were provided to speak to this issue. In SI Fig. 5b, we present the raw cell counts for colonies grown separately on 0.6% agar

plates for each strain under each condition. Comparison between the data in column 5 and column 6 is most relevant to the question raised by the reviewer. The red $\Delta vpsL$ strain occupies 50% of the final colony if it grows together with the cyan, isogenic $\Delta vpsL$ strain. The overall productivity of the colony is low in this case due to the absence of the osmotic pressure difference (see colony size measurement in SI Fig. 5a, fourth bar). When the same red $\Delta vpsL$ strain is co-inoculated with a cyan Rg_M (EPS⁺) strain, the relative frequency of the red $\Delta vpsL$ strain in the final colony is below 50% (Fig. 3b). However, the cell count of the entire colony is larger (SI Fig. 5a, third bar) in this case. The overall result is that the *absolute* cell count of the red $\Delta vpsL$ strain remains statistically the same when it is grown with a cyan $\Delta vpsL$ strain or with a cyan Rg_M strain. In the latter case, the $\Delta vpsL$ cells are located at the periphery, nonetheless, they do not grow more rapidly than the EPS⁺ cells which, embedded in an expanding polymer network, benefit from enhanced nutrient uptake due to the *local* osmotic differential.

Typos and the like:

(1) Data for Rg_M and DeltaBC in Supplementary Fig 3 are different from those in Figure 2, is this a different replica of the same experiments?

Yes. Fig. S3 (now new Fig. S4) shows a different set of replicas. We use the same batch of agar plates in each experiment to account for day-to-day variation in the agar plate production.

(2) Line 459 where -> were

Corrected as suggested.

(3) supplementary figure 2: using the same color code for Region I in panel b and c would help visually

Changed as suggested.

Referee #2:

The authors present work describing *Vibrio cholerae* colony biofilm formation and the connection between spreading of colonies on agar plates with osmotic pressure differences and the influence on nutrient uptake. The results are clear and the manuscript is easy to read. The motivation for the work and the findings extend from work by Brenner and coworkers on *Bacillus subtilis* (PNAS, 2012). The work presented in this submitted manuscript is solid and valuable for this understudied area of biofilm biology. It provides specific data to support important fundamental concepts and the figures are well designed and illustrate the science. Importantly, the authors include competition experiments wherein they demonstrate that matrix-producing cells outcompete non-matrix producers. Thus, the authors provide comprehensive data documenting the results of matrix production on spreading in *V. cholerae*, an organism that is studied widely by many laboratories. The impact of the work is very high and likely to be of broad interest.

The manuscript would be improved if the authors were clearer on the specific previous findings and hypotheses presented in the Brenner paper and those that are new here. The manuscript does briefly capture the overall nature of the 2012 paper but without any specifics and this gives the reader the sense that doing so could diminish the novelty of this manuscript. To the contrary, the clear statements of previous results and hypotheses would strengthen this manuscript and the generality of the findings and the growing sets of questions. The following specific comments involve these minor issues.

We appreciate the reviewer's support of our paper, as well as the very helpful comments for improvement. We apologize for not stating clearly enough the earlier contribution by the Brenner paper (Seminara *et al.*). We have now significantly improved the manuscript by being more specific about what was accomplished in the earlier Brenner paper and what is different in our present work. We refer to the Brenner work throughout our revised manuscript, and in particular, in our

revised introduction we lay out what Brenner and coworkers showed.

1. At the top of page 3, the authors state: “In biofilms, the high local concentration of polymer molecules surrounding the cells necessarily produces an osmotic pressure difference between the matrix and the external environment. It is not clear if or how such osmotic pressure gradients influence the mechanical properties of biofilms or the growth and fitness of the bacteria residing in them.” The first statement could be supported with a reference (Brenner or earlier reference). The osmotic pressure difference also depends, in principle, on the exact external environment (aqueous vs hypersaline vs sludge environments). Can the authors comment on this?

We have added the Brenner citation along with other additional references in the text as suggested by the reviewer. With respect to this question, indeed, the osmotic pressure differential could in principle depend on other external environmental factors. In the case of *Vibrio cholerae*, polymers present in the human gut, which have been shown to affect the mucus structure (*PNAS*, **113**, 7041-7046, 2016), could have similar osmotic effects on *V. cholerae* biofilms. In the environment, because *V. cholerae* transitions between fresh water and salty sea water, osmotic fluctuation could also affect biofilm formation and other physiological responses (*J. Bacteriol.* **191**, 4082-4096, 2013; *Environ. Microbiol.* **15**, 1387, 2013). A key difficulty in drawing a definite conclusion from studies of biofilms in these environments is to separate active gene-regulation-driven responses from passive osmotic responses of the matrix. We are currently working in this direction, with a focus on the response to mucus and its associated polymers. We have added relevant text both in the introduction and in the discussion.

2. In the following sentence, the authors accurately state: “Experimental and theoretical work by Seminara et al. analyzing *Bacillus subtilis* colonies on agar plates suggested a crucial role for EPS-generated osmotic pressure differences in facilitating nutrient uptake¹⁴.” However, the authors later state that: “We hypothesize that, relative to passive diffusion, an osmotic pressure-driven transport process could accelerate nutrient uptake from resources located at a distance from the biofilm cells, because transport times scale linearly with distance in osmotic-driven processes but quadratically in diffusion-limited processes²¹.” This notion of an osmotic pressure-driven transport process, however, is exactly what was concluded in the Brenner paper, wherein they discussed a result that “strongly suggests that the energetic investment implied in the production of EPS is rewarded by the consequent increase in nutrient uptake and results in a net fitness increase for the colony.” Their abstract even ended with the statement “the implications of this osmotically driven type of surface motility for nutrient uptake that may elucidate the reduced fitness of the matrix-deficient mutant strains.” The authors of this work on *V. cholerae* should specifically describe the major hypotheses and results that were published in the 2012 paper.

We thank for the reviewer for pointing out the need for clarification of this important issue. Indeed, in this particular paragraph in the manuscript, we are testing hypotheses similar to those proposed, but not yet confirmed, in the Brenner work. The Brenner manuscript showed a similar decreasing colony size trend with increasing agar concentration, however the Brenner manuscript did not explore whether this reduction could be attributed to changes in the mechanical properties of the agar. The experiments described in our present text paragraph and in Fig. 1e (colony growth on semi-permeable membrane) are designed to distinguish between changes in the mechanical environment and changes in osmotic pressure. Nonetheless, we have now modified the text here and in many other places to give appropriate credit to the pioneering work by Brenner and coworkers.

REVIEWERS' COMMENTS:

Reviewer #1 (Remarks to the Author):

The authors addressed all my curiosities and questions, and improved considerably all aspects that I wondered about in my previous review. I warmly recommend the manuscript for publication in Nature Communications.

Reviewer #2 (Remarks to the Author):

[No further comments for author.]